# *L*-Arginine prevents cereblon-mediated ubiquitination of glucokinase and stimulates glucose-6-phosphate production in pancreatic β-cells

Jaeyong Cho[1], Yukio Horikawa[2], Mayumi Enya[2], Jun Takeda[2], Yoichi Imai [3], Yumi Imai[4], Hiroshi Handa [5] & Takeshi Imai [1✉]

We sought to determine a mechanism by which *L*-arginine increases glucose-stimulated insulin secretion (GSIS) in β-cells by finding a protein with affinity to *L*-arginine using arginine-immobilized magnetic nanobeads technology. Glucokinase (GCK), the key regulator of GSIS and a disease-causing gene of maturity-onset diabetes of the young type 2 (MODY2), was found to bind *L*-arginine. *L*-Arginine stimulated production of glucose-6-phosphate (G6P) and induced insulin secretion. We analyzed glucokinase mutants and identified three glutamate residues that mediate binding to *L*-arginine. One MODY2 patient with GCK$^{E442*}$ demonstrated lower C-peptide-to-glucose ratio after arginine administration. In β-cell line, GCK$^{E442*}$ reduced *L*-arginine-induced insulin secretion compared with GCK$^{WT}$. In addition, we elucidated that the binding of arginine protects glucokinase from degradation by E3 ubiquitin ligase cereblon mediated ubiquitination. We conclude that *L*-arginine induces insulin secretion by increasing G6P production by glucokinase through direct stimulation and by prevention of degradation.

[1] Department Aging Intervention, National Center for Geriatrics and Gerontology, Obu, Aichi 474-8511, Japan. [2] Department of Diabetes and Endocrinology, Gifu University, Gifu, Gifu 501-1194, Japan. [3] Department of Hematology/Oncology, Research Hospital, Institute of Medical Science, University of Tokyo, Tokyo 108-8639, Japan. [4] Department of Internal Medicine, Fraternal Order of Eagles Diabetes Research Center, University of Iowa Carver College of Medicine, Iowa City, IA 52242, USA. [5] Department of Nanoparticle Translational Research, Tokyo Medical University, Shinjyuku, Tokyo 160-8402, Japan. ✉email: timai@ncgg.go.jp

Maturity-onset diabetes of the young (MODY) is the monogenic form of diabetes mellitus[1–3] of which MODY2 is caused by a mutation of glucokinase (GCK) gene. GCK is highly expressed in pancreatic β-cells and hepatocytes. It phosphorylates glucose in the presence of adenosine triphosphate (ATP) and magnesium[1–3]. The low affinity of GCK to glucose and minimum inhibition by its product glucose-6-phosphate (G6P) allow[4] GCK to increase G6P output efficiently in proportion to glucose input in β-cells[3]. G6P production initiated by GCK ultimately closes $K_{ATP}$ channel and induces $Ca^{2+}$ influx by opening the voltage-gated $Ca^{2+}$ channels leading to exocytosis of insulin granules in a glucose-dependent manner[5,6]. As expected, some GCK mutants that reduce kinase activity cause the impairment in glucose-induced insulin secretion (GSIS).

GCK produces G6P and adenosine diphosphate from glucose and ATP is in the presence of magnesium. In hepatocytes, GCK is masked by glucokinase regulatory protein (GKRP) and dissociation of GKRP through deacetylation by p300 triggers activation of GCK[7]. Although the regulation of GCK activity has been extensively studied in pancreatic β-cells[8–12], a binding molecule that regulates GCK activity has not been demonstrated in pancreatic β-cells. Here, we have identified that L-arginine binds GCK and regulates its activity and degradation.

L-Arginine is known to increase insulin secretion in the presence of glucose (ref. [1]). The widely accepted mechanism by which L-arginine increases insulin secretion is through cationic amino acid transporter 1–2 (CAT1-2, SLC7A1-2) that generate depolarizing current during uptake of arginine initiating $Ca^{2+}$ influx and insulin secretion in the presence of a permissive concentration of glucose[13–15]. Stereoisomer D-arginine stimulates insulin secretion without $Ca^{2+}$ influx[16,17] by interacting with UDP-glucose:glycoprotein glucosyltransferase 1 (UGGT1) in the endoplasmic reticulum (ER). Although stereospecific induction of insulin secretion by L-arginine is partly attributed to stereospecificity of CAT, there could be additional pathways that L-arginine utilizes to increase $Ca^{2+}$ influx in β-cells. We previously demonstrated that phosphofructokinase (PFK1 and PFK2) and hexokinase 1 (HXK1) in HeLa cell extract were L-arginine-binding proteins using L-arginine-immobilized magnetic nanobeads technology[18]. Here, we demonstrate that depletion of arginine reduces GSIS indicating that arginine is not a mere amplifier of GSIS but is required for GSIS. L-Arginine directly binds to GCK to stimulate G6P production. L-Arginine also prevents ubiquitination-dependent proteolysis of GCK in β-cells by the chemically inducible E3 ubiquitin ligase cereblon that is identified as a protein responsible for teratogenicity of thalidomide through ubiquitination-dependent degradation of cognate proteins[19–21].

## Results

### Glucokinase interacts with arginine in β-cells

To assess the extent by which arginine supports GSIS, NIT-1 β-cells were pre-cultured in arginine-depleted medium for 30 min. Thereafter, glucose was added to the medium and insulin secretion was analyzed (Fig. 1a). Depletion of arginine significantly blunted GSIS from NIT-1 β-cells. Interestingly, G6P administration induced dose-dependent insulin secretion both with and without arginine depletion (Fig. 1b). The increase of intracellular G6P by G6P administration was confirmed by mass spectrometry (Supplementary Fig. 1a, b). Thus, arginine is necessary for insulin secretion in response to glucose but not to G6P indicating that arginine may play a role for G6P production by GCK in β-cells.

We previously demonstrated that PFK1, PFK2, and HXK1 in HeLa cell extract were arginine-binding proteins using arginine-immobilized magnetic nanobeads technology, (Fig. 1c, Supplementary Fig. 1 and ref. [18]). Considering that PFK1/2 and HXK1

are enzymes of glycolysis[22,23], arginine might serve as a key regulator of glycolysis. Thus, we hypothesized that arginine also binds GCK and functions as an activator of GCK in β-cells. When tested using glucose- or arginine-immobilized magnetic nanobeads (Fig. 1c, Supplementary Fig. 1 and ref. [18]), GCK binds to both glucose and arginine (Fig. 1d). Next, we tested whether glucose and arginine compete for the binding of GCK to the immobilized magnetic nanobeads (Fig. 1e). GCK bound to immobilized glucose was eluted by glucose and arginine, although binding of GCK to immobilized arginine was competed less by glucose (Fig. 1e) indicating stronger binding of GCK to arginine than to glucose. Then, intracellular localization of GCK was analyzed with immunohistochemistry in NIT-1 cells (Fig. 1f). GCK signal was mainly merged with insulin in secretory vesicles and partially with proinsulin in the ER in agreement with the previous report[24,25].

### L-Arginine stimulates G6P production and insulin secretion

Figure 1 collectively indicates that arginine promotes GSIS through direct interaction with GCK. As a next step, we analyzed whether arginine alters GCK activity in NIT-1 β-cells. Administration of glucose or L-arginine to NIT-1 cells increased G6P but stereoisomer D-arginine fail to increase G6P (Fig. 2a). Interestingly, induction of insulin secretion as seen as difference in insulin secretion at 0.5–1.5 min at 7 mmol/l glucose is faster after L-arginine administration compared with D-arginine (Fig. 2b). L-Arginine stimulated insulin secretion stronger than D-arginine in the presence of glucose but not in the absence of glucose (no-arginine, and Fig. 2c, d). GCK bound to L-arginine was eluted by L-arginine but not by D-arginine indicating that only L-arginine interacts with GCK (Fig. 2e). Taken together, L-arginine stereospecifically binds GCK, increases G6P contents, and promotes insulin secretion. Of note, glucose-independent insulin secretion can be stimulated by both L- and D-arginine, which is likely due to UGGT1, another arginine-binding protein that binds to both L- and D-arginine (Supplementary Fig. 2a–d)[26].

### Three glutamate residues are involved in arginine signaling

To identify the residues within GCK that are required for binding to L-arginine and insulin secretion stimulated by L-arginine, five mutants of GCK (Y214C, E256K, A456V, E157T, and E443Δ) were prepared and analyzed (Fig. 3 and Supplementary Fig. 3). Residues were chosen based on MODY2 patient mutations[11,26–30] and our previous paper showing the interaction of cationic amino acid arginine with acidic glutamate residue[26]. Wild-type and E443Δ GCK bound L-arginine strongly, however, binding to L-arginine was hardly observed for the E256K and E157T GCK mutants (Fig. 3a and Supplementary Fig. 3a, b). When insulin secretion in response to L-arginine was compared for all five mutants, insulin secretion was impaired for three GCK mutants, namely E256K, A456V, and E443Δ (Fig. 3b). Thus, the E256 residue appears to be critical for both L-arginine binding and L-arginine-induced insulin secretion. On the other hand, the E443 residue is necessary for L-arginine-induced insulin secretion but not for binding of L-arginine to GCK. Although E443Δ was not sufficient to prevent binding of L-arginine to GCK, we noted that there is a cluster of E residues (E442, and E443) in the C-terminal region of GCK[26]. Using GCK structure data (ref. [31], PDB-1V4S, and 1V4T), we predicted the three dimensional position of these three E residues (E256, E442, and E443) on the GCK protein (Supplementary Fig. 3c, d) and found that these residues creates a gate-like structure that can serve as a binding pocket for L-arginine. Therefore, we made a triple mutant E256/442/443R (Fig. 3e–g) and found that this GCK mutant showed a marked impairment in arginine binding (Fig. 3e), the increase in G6P by L-arginine (Fig. 3f), and arginine-induced insulin secretion

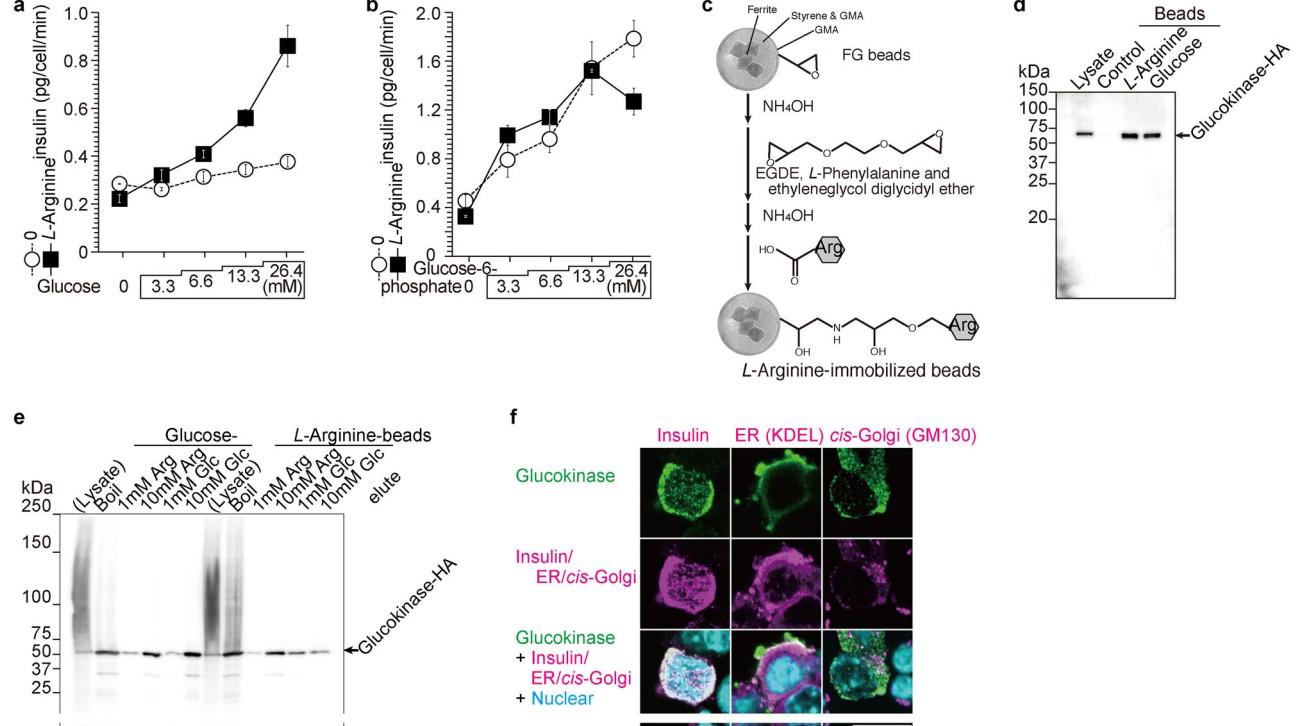

**Fig. 1 Arginine binds to glucokinase. a, b** Glucose-6-phosphate (G6P) rescued the reduction of insulin secretion in the absence of *L*-arginine in NIT-1 cells. In the presence of *L*-arginine, glucose and G6P induced insulin secretion. In the absence of *L*-arginine, only G6P (open circle in **b**), but not glucose (open circle in **a**) induced insulin secretion. Data represent the mean ± S.E. (*n* = 4). **c** Preparation of arginine-immobilized magnetic nanobeads[18]. Epoxy groups on FG beads were aminolyzed by $NH_4OH$ and coupled to EGDE. Epoxy groups were re-aminolyzed by $NH_4OH$. The beads were then coupled with carboxyl groups of *L*-arginine in DMF containing EDC, trimethylamine and DMAP (Supplementary Fig. 1). **d** Glucokinase binds to *L*-arginine and glucose directly. Glucokinase produced in HEK293 cell transfected with glucokinase-HA expression vector were mixed with arginine or glucose-immobilized beads using beads without arginine or glucose as negative control. Both *L*-arginine and glucose bind glucokinase-HA. **e** Glucokinase-HA bound to immobilized glucose was eluted by *L*-arginine and glucose. Glucokinase-HA bound to *L*-arginine beads was eluted by *L*-arginine but not by glucose. **f** Glucokinase co-localizes with insulin in the secretory granule and not in the endoplasmic reticulum (ER) and the Golgi apparatus in NIT-1 cells. Glucokinase (green), insulin (magenta), ER (magenta, KDEL), *cis*-Golgi (magenta, GM130), and nucleolus (blue) were stained and merged pictures were shown at the bottom. The glucokinase showed high colocalization with insulin and colocalized less with ER and Golgi network indicating that glucokinase mainly locates in secretory vesicles. Scale bar 20 μm.

(Fig. 3g). Taken together, E256, E442, and E443 residues of GCK are involved in *L*-arginine binding and insulin secretion.

**Impaired insulin secretion in E442* MODY2 patients**. We screened 489 Japanese MODY patients for the GCK variants that involve E442 and E443 and identified a subject with variant GCK[E442*] (Supplementary Figs. 3a, S4a and refs. [27–29]). When a subject with GCK[E442*] variant (Fig. 4a, b and Supplementary Fig. 4b) and two other healthy subjects were tested, the E442* subject exhibited a lower value for the homeostatic model assessment of β-cell (Fig. 4a). The subject with GCK[E442*] variant displayed lower insulin secretion during fasting. In an arginine tolerance test, the E442* variant showed a lower C-peptide-to-glucose ratio (ref. [32] and Fig. 4b), indicating impaired insulin secretion in response to arginine. When expressed in NIT-1 cells, E442* and the similar E442/443R GCK mutants showed reduced arginine-induced insulin secretion (Fig. 4c), GSIS (Supplementary Fig. 4c), and G6P levels in the presence of arginine (Fig. 4d). Our data from the GCK[E442*] subject and similar mutants studied in NIT-1 cells suggested that *L*-arginine modulates GCK-dependent insulin secretion, which was not apparent from a group of mutations studied by Pueyo's study[16].

**Arginine protect GCK from ubiquitination and degradation**. Next, we addressed a mechanism by which binding of arginine to

GCK increases G6P levels in β-cells. We noted that GCK has the ubiquitin interacting motif (UIM[33,34]) at the C-terminal[35,36] near E442 residue. We did not observe any difference in total ubiquitinated proteins in lysates in the presence or absence of arginine when ubiquitinated proteins were detected using ubiquitin-Myc expression vector (Fig. 5a). However, ubiquitinated GCK was observed only after arginine depletion (Fig. 5a) indicating that arginine prevents ubiquitination of GCK. Considering that the blood arginine level decreases during (long) fasting, the release of arginine from GCK may increase ubiquitin-mediated proteolysis of GCK when glucose availability and requirement for insulin secretion both are reduced during fasting. Twelve-hour treatment of NIT-1 cells by proteasome inhibitor MG132 increased WT GCK level approximately ten-fold but did not change the level of E256K mutant protein (Fig. 5b). Insulin secretion induced by *L*-arginine was also increased in the presence of MG132 in NIT-1 cells expressing WT GCK but not those expressing E256K (Supplementary Fig. 5a). The data indicated that ~90% of WT GCK is degraded in 24 h supporting the hypothesis that arginine depletion accelerates degradation of GCK through ubiquitination. However, it appears that the presence of low levels of GCK is sufficient to increase insulin secretion in response to *L*-arginine indicating that direct interaction between *L*-arginine and GCK acutely increases GSIS (Supplementary Fig. 5b, c). Next, to address how ubiquitination

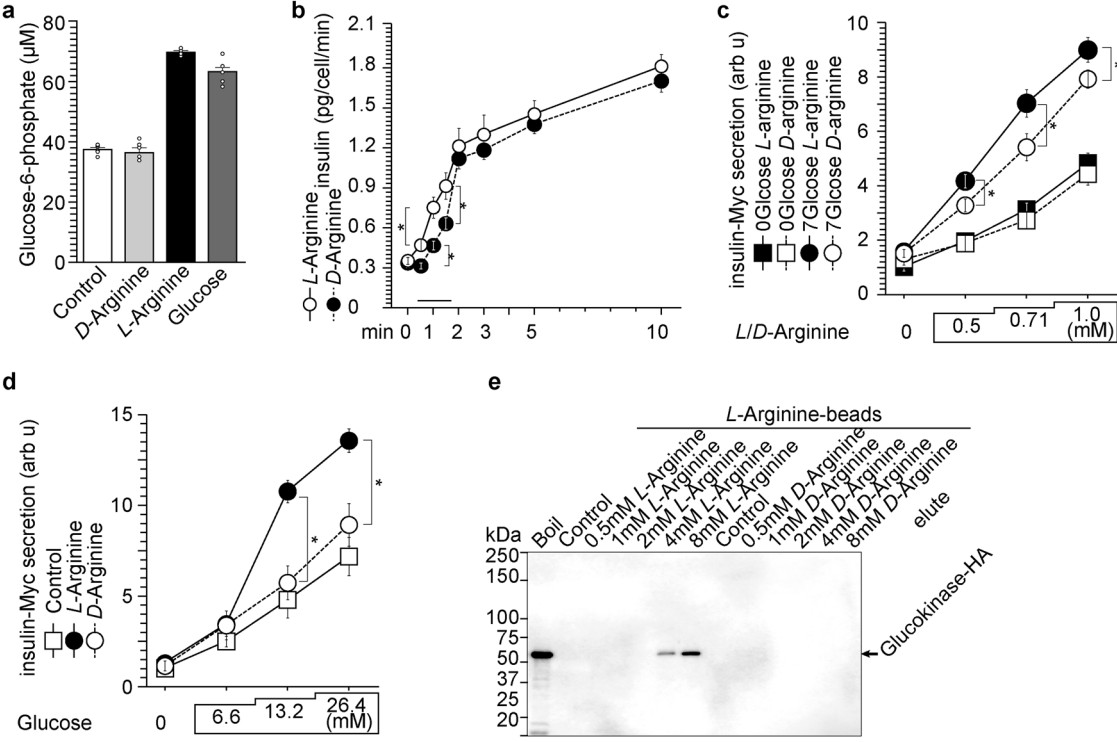

**Fig. 2 *L*-Arginine stimulates glucose-6-phosphate (G6P) production and insulin secretion. a** *L*-Arginine (2 mmol/l) and glucose (7 mmol/l) stimulated G6P production in NIT-1 β-cells, but stereoisomer *D*-arginine (2 mmol/l) did not induce G6P production. Data represent the mean ± S.E. (*n* = 6). **b** One mmol/l *L*-arginine-induced insulin secretion is slightly more than one mmol/l *D*-arginine does in NIT-1 β-cells. *L*- or *D*-arginine were administered to *L*-arginine depleted NIT-1 cells. Data represent the mean ± S.E. (*n* = 3). *$p < 0.05$ (*n* = 6). **c** In the presence of glucose, *L*-arginine stimulated insulin secretion more than *D*-arginine. In the absence of glucose, both *L*- and *D*-arginine stimulated insulin secretion similarly. Data represent the mean ± S.E. (*n* = 3). *$p < 0.05$ (*n* = 6). **d** One mmol/l *L*-arginine augments glucose-dependent insulin secretion. Data represent the mean ± S.E. (*n* = 3). *$p < 0.05$ (*n* = 6). **e** Glucokinase bound to *L*-arginine but did not bind to *D*-arginine. Of note, UGGT1 binds to *L*- and *D*-arginine (Supplementary Fig. 2a and ref. [26]).

of GCK is accelerated in the absence of arginine, we tested the role of cereblon, the only E3 ubiquitin ligase inducible by the drug (thalidomide and its analogues, immunomodulatory imide drugs) among more than 500–1000 E3 ubiquitin ligases in humans[19–21]. In HEK293 cells in which cereblon was knocked out, we found that cereblon is indispensable to observe ubiquitination of GCK (Fig. 5c and Supplementary Fig. 5d). Using mutagenesis analysis, two lysine residues of K458 and K459 are shown to be responsible for cereblon-dependent ubiquitination that is reduced by arginine (Fig. 5d). K458/459R mutation prevented ubiquitination in the absence of arginine indicating that these residues are responsible for arginine-dependent ubiquitination out of 26 lysine residues in GCK (Fig. 5d). When HEK293 cells were transfected with GCK, ubiquitin, and cereblon expression vectors, both expression of GCK determined by Western blot and G6P contents decreased over 4 h under arginine depletion in WT expressing cells while both levels remained unchanged in E256K expressing cells (Fig. 5e, f). In addition, *L*-arginine was shown to reduce the interaction between GCK and cereblon (Fig. 5g). Moreover, *L*-arginine stimulates GCK activity in vitro (Fig. 5h and Supplementary Fig. 5e). Taken together, it appears that cereblon binds to GCK in the absence of arginine and ubiquitinates GCK, while arginine administration dissociates GCK from cereblon and promote phosphorylation of glucose by GCK to induce insulin secretion (Supplementary Fig. 5f).

## Discussion

In fed state, *L*-arginine binds to GCK via three glutamate residues of E256, E442, and E443, stimulates G6P production, and promotes insulin secretion. During fasting, when arginine levels reduce, GCK depleted of arginine gets ubiquitinated by cereblon and degrades. As a MODY2 patient with E442* failed to increase insulin secretion in response to arginine, *L*-arginine mimetics will be a candidate to improve insulin secretion in a certain MODY. Proteolysis targeting chimeras technology allowed us to identify that the establish combined chemical is responsible for degradation of mutated GCK specifically.

Arginine is one of the semi-essential amino acids obtained mainly by food intake. GCK structurally exists in two forms, one being the active form, and the other the inactive form (ref. [31] and Fig. 3c, d). In the fed state, when the glucose and *L*-arginine availability is high, we propose that *L*-arginine binds to GCK via at least three E residues, E256/E442/E443, stimulates its hexokinase activity toward G6P production, and induces insulin secretion (ref. [31] and Fig. 3c, d). In the fasting state when *L*-arginine and glucose concentrations are lower, they dissociate from GCK reducing G6P production ([31], and Fig. 3d) and increasing the degradation of GCK by cereblon.

E442* MODY2 patient exhibited impaired insulin secretion after fasting and during arginine tolerance test. Consistent with this, β-cells expressing the mutant GCK showed minimal *L*-arginine binding activity (Fig. 3a and Supplementary Fig. 3a, b) and *L*-arginine-induced insulin secretion (Figs. 3b, 4c). Combining data from various mutant studies, we propose that E256/E442/E443 of GCK are involved in *L*-arginine-binding as well as *L*-arginine-induced insulin secretion, providing a molecular mechanism by which E256K and E442* mutations lead to MODY2. *L*-Arginine has higher affinity to GCK than glucose (Fig. 1e) considering that *L*-arginine competes glucose–GCK

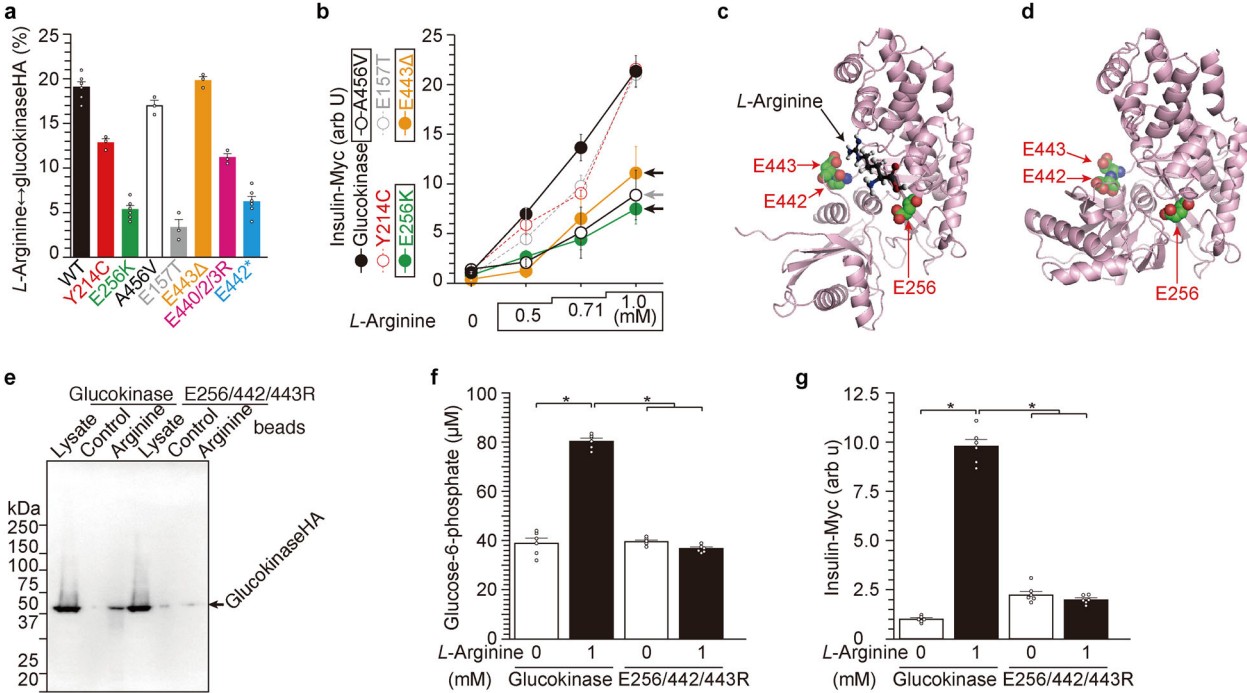

**Fig. 3 E256, E442, and E443 residues are involved in arginine-induced insulin secretion in NIT-1 β-cells. a** Glucokinase mutants E256K, E440/442/3R, and E442* showed reduced binding to L-arginine-immobilized beads when several glucokinase proteins were analyzed (Supplementary Fig. 2b). **b** NIT-1 cells expressing E256K, E443Δ, and A456V-glucokinase mutants showed reduced insulin secretion in response to L-arginine compared with those expressing WT glucokinase. Data represent the mean ± S.E. (n = 3, **a** and **b**). **c**, **d** Three glutamate residues of E256, E442, and E443 of glucokinase are located closely and form a gate-like structure for the binding of L-arginine based on the glucokinase protein structural data (PDB-1V4S[31]). Impaired arginine-binding (**e**), G6P production (**f**), and insulin secretion (**g**) in NIT-1 cells expressing E256/442/443R mutant glucokinase. Data represent the mean ± S.E. (n = 3). *p < 0.05 (n = 6, **f** and **g**).

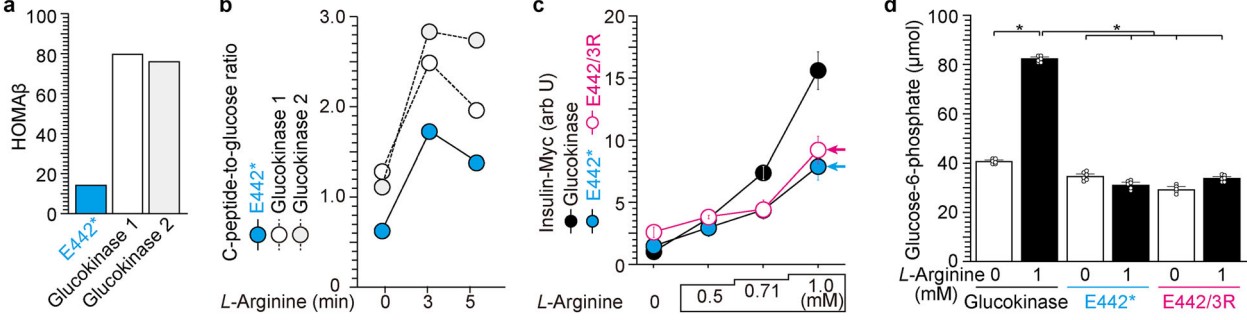

**Fig. 4 Impaired insulin secretion in a subject with glucokinase^E442*.** Homeostatic model assessment of β-cell (HOMAβ, **a**) and C-peptide-to-glucose ratio (**b**) were reduced in a subject with E442* glucokinase mutation compared with two controls with WT glucokinase. Lower HOMAβ indicates reduced ability to secrete insulin after fasting state (**a**). The C-peptide-to-glucose ratio is one of the markers demonstrating efficiency of insulin secretion[32] (**b**). **c** L-arginine-induced insulin secretion was reduced in NIT-1 cells that expressed E442* and E442/3R mutant glucokinase by transfecting with pCDNA-GCK-HA and insulin expression vectors. **d** G6P production in response to L-arginine was reduced in NIT-1 cells that expressed E442* and E442/3R mutant glucokinase by transfecting with pCDNA-GCK-HA and insulin vectors. Data represent the mean ± S.E. (n = 3). *p < 0.05 (n = 6).

complex (Fig. 1e). These data suggested that arginine-based drug might have higher affinity to GCK and can be utilized to increase insulin secretion.

Stereoisomer D-arginine also stimulates insulin secretion but less potently when compared with L-arginine[17,30]. This may be due to D-arginine administration being accompanied by less Ca$^{2+}$ influx than L-arginine when insulin secretion is promoted from the secretory granule[17,30]. In addition, we recently reported that L- and D-arginine has a shared target in the ER for regulating insulin secretion (Supplementary Fig. 2 and ref. [26]). Here, we show evidence that GCK is an L-arginine-preferential target at the secretory granules by which L-arginine promotes insulin

secretion. Thus, arginine does appear to act via at least two signaling pathways, one responsive to both L-and D- arginine in the ER[26] and the other only responsive to L-arginine at the secretory granules, both synergistically act to stimulate insulin secretion.

During fasting when circulating glucose and arginine levels are lower, insulin level needs to be lowered as well to prevent hypoglycemia. Considering that GCK plays a key role in initiating GSIS, degradation of GCK by ubiquitination under arginine deprivation may serve as one of multiple mechanisms to prevent hypoglycemia during fasting[1]. In the current study, we identified UIM[33–36] at C-terminal of GCK as a target of ubiquitination by E3 ubiquitin ligase cereblon whose binding to GCK is regulated

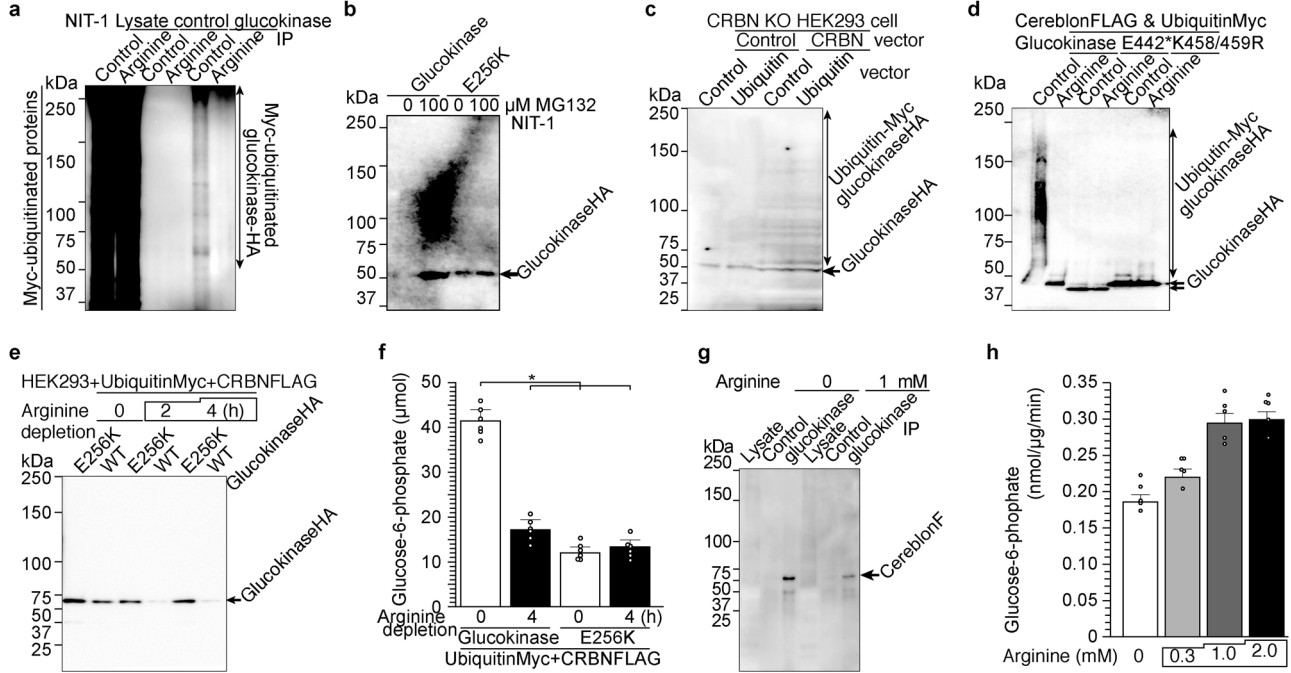

**Fig. 5 L-Arginine prevents ubiquitination of glucokinase and stimulates insulin secretion. a** Arginine inhibits ubiquitination of glucokinase. NIT-1 cells were transfected with ubiquitin-Myc expression vector and subjected to arginine depletion. Lysate, control, and glucokinase-IPed extracts were visualized with anti-Myc antibody. **b** Twenty-four hour incubation with proteasome inhibitor MG132 prevented reduction of WT glucokinase, but had little effect on the expression and ubiquitination level of E256K mutant glucokinase. **c** Ubiquitination of the glucokinase-HA is seen in WT HEK293 cells, but not in cereblon KO HEK293 cells. The reduction of cereblon in cereblon KO cells is shown in Supplementary Fig. 5a. **d** K458 and K459 residues of glucokinase are required for arginine-reducible and cereblon-dependent ubiquitination in the absence of arginine. NIT-1 cells transfected with WT, E442*, or K458/459R, glucokinase expression vectors along with cereblon-FLAG and ubiquitin-Myc expression vectors were incubated in arginine-depleted medium. Arginine depletion reduced WT glucokinase protein (**e**) and activity (**f**). Data represent the mean ± S.E. ($n = 3$). *$p < 0.05$ ($n = 6$). **g** Arginine inhibits interaction of glucokinase and cereblon. **h** Arginine increases G6P production by recombinant glucokinase protein in cell free system. ($n = 6$).

by L-arginine, thus providing a molecular mechanism by which GCK availability is regulated during fasting.

In summary, we combined studies performed in a cell free system based on magnetic nanobeads, NIT-1 cells, and MODY patients to unmask a previously unrecognized target of L-arginine in β-cells. We propose a novel pathway by which L-arginine stereoisomer regulates GSIS through GCK in β-cells that may underline pathology of some cases of MODY2 that involve amino acid residues important for L-arginine interaction.

## Methods

**Antibodies and cell culture**. The following antibodies were purchased: insulin (L6B10, Cell Signaling Technology, Danvers, MA, USA and sc-9168, Santa Cruz, California CA, USA), GM130 (cis-Golgi network staining, 610822, BD Transduction Laboratories, New York, NY, USA), FLAG (F-1804, Sigma, St Louis, MO, USA), HA (anti-HA high affinity 3F10, Roche, Basel, Switzerland), KDEL (ER staining, SPA-827, Stressgen, Victoria, BC, Canada), and Alexa Fluor-conjugated secondary antibodies (Thermo Fisher, Rockford, IL, USA). Mouse pancreas-derived NIT-1 cells (purchased from ATCC® CRL-2055™, Manassas, VA, USA) were maintained as described previously[18]. Briefly, the cells were plated at a density of $1.5–3.0 \times 10^6$ cells/60-mm dish, and the medium (F-12K, Kaighn's Modification of Ham's F-12 Medium, 2 mmol/l L-arginine and 7 mmol/l glucose) after 48 h of culture medium was exchanged to fresh medium after 48 h of culture.

**Insulin secretion analysis**. Insulin secretion activity was determined using a commercial ELISA kit (Shibayagi, Gunma, Japan) as described previously[18,26]. Intracellular L-arginine contents are as follows; control cells are 0.831 ± 0.16 (f mol/cell), 30 m L-arginine depleted cells are 0.091 ± 0.08 (f mol/cell)[37,38].

**Imaging study**. Immunocytochemistry was analyzed as described[26,39].

**Glucokinase expression in HEK293 cells**. pCDNA-GCK-HA and ubiquitin-Myc (Fig. 5c, d) expression vectors were transfected to HEK293FT cells using Lipofectamine® 2000 (Thermo Fisher). Transfection efficiency was ~90%. The cells were

cultivated in DMEM and lysed in buffer containing 20 mmol/l HEPES-NaOH (pH 7.9), 1 mmol/l MgCl₂, 0.2 mmol/l CaCl₂, 100 mmol/l KCl, 0.2 mmol/l EDTA, 10% Glycerol, 0.1% Nonidet P-40, 1 mmol/l Dithiothreitol, 0.2 mmol/l Phenylmethylsulfonyl fluoride (PMSF, Nacalai Tesque, Kyoto, Japan), 3% n-Octyl-β-D-glucoside (DOJINDO, Tokyo, Japan). The lysate was mixed with arginine/glucose-immobilized magnetic nanobeads (Fig. 1e, f), or directly analyzed by SDS-PAGE and Western blotting with HA or GCK antibody.

**Glucose-6-phosphate (G6P) production analysis**. NIT-1 cells were incubated with L-arginine- and glucose-free F-12K for 30 min followed by administration of 7 mmol/l glucose, 1 mmol/l L- or D-arginine for 30 min. Subsequently, the cell lysate was collected and G6P contents were measured using G6P contents were determined using a commercial kit (Glucose-6-Phosphate Fluorometric Assay Kit No. 700750, Cayman chemical, Ann Arbor, Michigan, USA). We analyzed intracellular G6P concentration with metabolome analysis[37,38].

**L-Arginine treatment of NIT-1 cells**. NIT-1 cells were cultured in F-12K medium (2 mmol/l L-arginine) with 10% FCS. Briefly, the cells were treated with L-arginine-free F-12K for 30 min followed by addition of a final concentration of 1 mmol/l L-arginine for 5 min or the indicated duration.

**Glucokinase expression**. Mutant GCK-HA was introduced to pCDNA3 expression vector using mutated primers (Supplementary Fig. 2h). Insulin-Myc and these mutated/WT GCK-HA expression vectors were transfected to NIT-1 cells and secreted insulin was measured with ELISA kit. For arginine/glucose binding assay, the lysate was mixed with arginine- or glucose-immobilized magnetic nanobeads and then eluted GCK-HA was analyzed by Western blotting. E. coli expressed recombinant GCK protein was purchased from Abcam (ab123840).

**In vitro kinase assay**. Recombinant GCK protein (0.1 μg) was mixed with kinase buffer (25 mM HEPES pH 7.5, 2 mM MgCl₂, 0.01 mM DTT, 2 mM ATP, 5 mM glucose, 0.1 mg/mL BSA), and was incubated for 60 min at 37 °C. G6P was determined using two commercial kits (Glucose-6-phosphate fluorometric assay kit No. 700750, Cayman chemical, Ann Arbor, Michigan, USA, and universal kinase assay kit AU-AB138879, Abcam).

**Immunoprecipitation and Western blot**. Immunoprecipitation and Western blot analysis were performed as described previously with minor modifications[18,40]. NIT-1 cells transfected with pCDNA-insulin-Myc expression vector were cultivated in arginine-free medium for 30 min before arginine was complemented. Cells were lysed in buffer containing 20 mmol/l HEPES-NaOH (pH 7.9), 1 mmol/l $MgCl_2$, 0.2 mmol/l $CaCl_2$, 100 mmol/l KCl, 0.2 mmol/l EDTA, 10% glycerol, 0.1% Nonidet P-40, 1 mmol/l Dithiothreitol, 0.2 mmol/l Phenylmethylsulfonyl fluoride (PMSF, Nacalai Tesque, Kyoto, Japan), 3% n-Octyl-β-D-glucoside (DOJINDO, Tokyo, Japan). After incubation on ice for 30 min, lysates were centrifuged at $12,000 \times g$ for 15 min at 4 °C and dialyzed against lysis buffer without n-Octyl-β-D-glucoside for 3 h at 4 °C. After dialysis, the supernatant was incubated with the indicated antibody for 120 min at 4 °C. The samples were then incubated with Protein G Sepharose 4 Fast Flow (GE Healthcare). After an additional washing, the precipitates were heat-denatured in SDS-sample buffer. For immunoblot analysis, proteins were separated by SDS/PAGE and transferred to a PVDF membrane for Western Blot. The membranes were blocked with 5% nonfat milk in Tris-buffered saline with Tween 0.1%, incubated overnight with the mixture primary antibody (c-Myc [1:1000]) in the presence of Can Get Signal solution (TOYOBO, Osaka, Japan) at 4 °C, washed, incubated with the secondary antibody for 60 min at room temperature, and then washed again. Immune-complexes were detected using Immunostar LD (Wako, Osaka, Japan) substrate. Signals were quantified with the LAS 4000 imaging system (GE Healthcare, Chicago, IL). ImageJ was used for densitometry of scanned membranes[18,40].

**Human MODY study**. Arginine tolerance test was performed in two MODY2 subjects at the Gifu University Hospital in Gifu, Japan. Blood samples were obtained via a catheter inserted into a superficial forearm vein. In total, 5 g arginine as 10% arginine hydrochloride was administered intravenously over 30 s, the end of which was designated as time point 0. Blood samples were obtained at time 3 and 5 min for the determination of insulin levels.

**Ethical approval**. The clinical study of arginine tolerance test protocol was approved by the Institutional Review Board of Gifu University (Gifu-No. 25–153). Written informed consent was obtained from all participants[17–19].

**Statistics and reproducibility**. The values are reported as the means ± standard error ($n = 3$–6). Statistical significances (single-sided the Student's $t$ test or one-way/two-way analysis of variance) are indicated in figure legends as follows: *$p < 0.05$. $n = 6$ for statistical analysis[18,40]. For reproducibility of key experiments, we performed total number of experiments exceed ten times including experiments for setting conditions with similar results. Also, we employed multiple approaches to confirm one result.

**Reporting summary**. Further information on research design is available in the Nature Research Reporting Summary linked to this article.

## Data availability

The authors declare that all reported data in the main and supplementary files will be provided to other investigators as requested. Such requests should be addressed to T.I. (timai@ncgg.go.jp). Full blots are shown in Supplementary Data 1. Source data underlying plots shown in figures are provided in Supplementary Data 2.

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

## Acknowledgements

We are grateful to members of our department at the National Center for Geriatrics and Gerontology (NCGG) for their helpful discussions, and Dr. N. Maekawa, M. Umeda, and Y. Tsugawa for technical assistance. We thank Mr. Joseph A. Promes of University of Iowa for critical reading of the manuscript. This work was supported by a Grant-in-Aid from the Ministry of Education, Culture, Sports, Science and Technology (MEXT 18659493), the Japan Science and Technology Agency (A-STEP-AS2312036G and FY2013-SICP) and NCGG (28–25, 19–27) to T.I., and MEXT-Supported Program for the Strategic Research Foundation at Private Universities S1411011 to H.H. Yu I. is financially supported by National Institutes of Health, USA to Y.I. (R01-DK090490) and American Diabetes Association to Y.I. (1–17-IBS-132).

## Author contributions

J.C., Y.H., M.E., J.T., and H.H. performed the experiments, Y.H. and T.I. designed the experiments, analyzed the data, and T.I. wrote the manuscript. Yo I. and Yu I. interpreted the data and wrote the manuscript.

## Competing interests

The authors declare no competing interests.
