## [Peer Review File · Communications Biology]

Reviewers' comments:

Reviewer #1 (Remarks to the Author):

In this manuscript the authors show that glucokinase is a L-Arginine-specific target protein. In Nit-1 beta cells, three glucokinase mutants (E256, E442 and E2443) were identified to be relevant for L-Arginine binding and L-Arginine-induced insulin secretion. In addition, it is demonstrated that L-Arginine prevents ubiquitination-dependent proteolysis of glucokinase by the chemical inducible E3 ubiquitin ligase cereblon. The authors also give a link to a MODY2 patient carrying E442 mutation in the glucokinase showing lower insulin secretion in response to L-Arginine.

The manuscript deals with an interesting and novel research question. The topic is considered as of importance for its field, however, there are several major concerns of the manuscript, that make it difficult for publication in its current form.

Major points:

1. Most experiments have been repeated for only three times. For immunoblots the numbers of independent experiments is not mentioned. According to the standards of the British Journal of Pharmacology five is a minimum of numbers needed for reliable statistics (Curtis, Alexander et al. 2018). The authors have to increase their n numbers throughout the whole manuscript, human studies might be excluded due to the lack of patients. In addition, with reference to several guide lines it is more transparent to just plot the independent data points for the different measurements (Vaux 2012). The authors have to change this throughout the manuscript.
2. The manuscript needs critical reading. Several sentences in the Manuscript and the Figure legends are incorrect or words appear twice (e.g. Glucokinase is interacts with arginine in beta cells – beginning of the results part; Figure 2c In the presence of glucose, L-Arginine more stimulates insulin secretion more than D-arginine).
3. In some Figures the unit for secreted insulin is not given. Please indicate the unit of the measured insulin (e.g. pg/cell/min) or if and how the data has been normalized (e.g. Figure 1a,b, Figure 2b).
4. In many figures it is a challenge for the reader to recognize what exactly was measured. The axis labeling is often lost in the legends of the curves presented in the figures. The axis labeling should be clearly visible (e.g. Figure 1a,b). The labeling for curve legends should be shown above or besides the curves or columns but not along the y-axis labeling.

Minor points:

1. In figure legend 3 last sentence: f and g do not exist.

2. Why are p-values and Asterix mentioned in some figures (Figure 2c,d). It is difficult to see which comparisons shall be indicated with the individual p-values.

3. Spelling should be consistent throughout the manuscript L-Arginine or L-arginine

Curtis, M. J., S. Alexander, G. Cirino, J. R. Docherty, C. H. George, M. A. Giembycz, D. Hoyer, P. A. Insel, A. A. Izzo, Y. Ji, D. J. MacEwan, C. G. Sobey, S. C. Stanford, M. M. Teixeira, S. Wonnacott and A. Ahluwalia (2018). "Experimental design and analysis and their reporting II: updated and simplified guidance for authors and peer reviewers." *Br J Pharmacol* 175(7): 987-993.

Vaux, D. L. (2012). "Know when your numbers are significant." *Nature* 492(7428): 180-181.

Reviewer #2 (Remarks to the Author):

The hypothesis tested in this study is good and interesting but there are several improvements that have to be made in this manuscript. It would be fruitful for this manuscript to improved discussion with more authentic data rather than citing unpublished (submitted) manuscripts/data. It is also encouraged to cite relevant literature while writing methods section for authenticity of your research work. It would be highly appreciable, if two pics (f and g) from Fig. S3 be included in the main manuscript. Supplementary information Fig. S4 has not been referred anywhere in the manuscript. Please see the attached file for details.

Reviewer #3 (Remarks to the Author):

In this study the authors describe that L-arginine (2 mM) increases glucose-6-phosphate generation and glucokinase activity as well as glucose-induced insulin secretion through interaction and activation of glucokinase in the NIT-1 beta cell line. They propose that this effect is achieved through interaction of L-arginine with three specific amino acids in the glucokinase protein. This effect is absent in a glucokinase mutant which causes in patients MODY2.

General comment: The effects are of interest but they require further experiments for purposes of support as detailed below.

However, it is very irritating that the authors have submitted a manuscript which is rather a draft but not a manuscript in a proper final form. In many locations in the text on pages 14-15,

there are still "XXX" symbols in the text rather than proper information. In addition the English language requires intensive revision.

Specific comments:

1. Additional information and support must be through the following additional experiments, to provide additional support for their conclusions:

a) Measurement of glucose-dependent glucokinase activity in the absence and presence of 1 mM and 2 mM L-arginine in vitro with isolated recombinant human glucokinase.

b) Identical experiments with a cytosolic glucokinase extract from NIT-1 cells.

2. Mention the arginine concentration at the beginning of the results text and in the legends to the figures.

3. Explain "cereblon" in more detail and with reference.

4. It is well-known that beta cell glucokinase enzyme activity is increased through activation by interaction with PFK2 (such aspects of other established activators of glucokinase are not mentioned in this MS). Therefore the question: might arginine increase glucokinase activity in some way through interaction with the binding of PFK2 to glucokinase ?

5. In liver GRP (glucokinase regulatory protein) is a strong inhibitor of GK activity.

Is there evidence for interaction of arginine with GRP binding to GK ?

6. Fig. 1: It is irritating and cannot be explained why the authors observed in Fig. 1b, that G6P but not glucose induced insulin secretion from NIT-1 cells. Rather the opposite should be the case, because glucose but not the phosphorylated G6P can enter an intact cell. This result is completely implausible.

7. Can arginine immobilized with magnetic beads enter intact cells ? Where is this documented ?

8. It is not correct when the authors state in legend to Fig. 1 that GK is localized in secretory vesicles. This statement completely obscure. The reviewer does not understand how such a statement in view of the general knowledge about GK location in the literature can be made.

Response to reviewers

Hereafter our answer of your previous email, “Reviewer Assessment Report” and manuscript is written in green letters.

Dear Prof Imai,

Your manuscript entitled "*L*-Arginine releases glucokinase from cereblon-mediated ubiquitination and stimulates glucose-6-phosphate production in pancreatic beta cells" has now been seen by 3 referees. You will see from their comments below that while they find your work of interest, some important points are raised. We are interested in the possibility of publishing your study in Communications Biology, but would like to consider your response to these concerns in the form of a revised manuscript before we make a final decision on publication.

To address the reviewers' concern, we ask you to test whether *L*-arginine is sufficient to activate glucokinase in vitro and to test whether arginine's effects on glucokinase are indirect, as suggested by reviewer #3. We also ask you to please go over this manuscript carefully, correcting all the errors, providing more experimental details, and citing relevant references. Copy-editing of this manuscript is recommended as well.

We therefore invite you to revise and resubmit your manuscript, taking into account the points raised. Please highlight all changes in the manuscript text file.

*Thank you very much for considering our manuscripts for publication and providing encouraging review. I made change in green letters below and in the revised manuscript.

We are committed to providing a fair and constructive peer-review process. Do not hesitate to contact us if you wish to discuss the revision in more detail or if there are specific requests from the reviewers that you believe are technically impossible or unlikely to yield a meaningful outcome.

At the same time, we ask that you ensure your manuscript complies with our editorial policies. Please see our revision file checklist for guidance on formatting the manuscript and complying with our policies. You will also find guidelines on structuring the response to reviewers document.

Please use the following link to submit your revised manuscript, point-by-point response to the referees' comments (which should be in a separate document to any cover letter) and any additional files:

<https://mts-commsbio.nature.com/cgi-bin/main.plex?el=A3Cx5Bcn4A3ZON2I4A9ftdSm0nDJcF1GkEH3gExe0AZ>

We hope to receive your revised paper within three months. We understand that due to the current global situation, the time required for revision may be longer than usual, and we are happy to accommodate this as needed.

Please do not hesitate to contact me if you have any questions or would like to discuss these revisions further. We look forward to seeing the revised manuscript and thank you for the opportunity to review your work.

Best regards,

Jung-Eun Lee, PhD
Associate Editor, Communications Biology
One New York Plaza, Suite 4600
New York, NY 10004-1562
orcid.org/0000-0003-0184-3440
jung-eun.lee@nature.com

Referee expertise:

Referee #1: Regulation of insulin secretion by *L*-Arg

Referee #2: Biochemistry and diabetes

Referee #3: Regulation of glucokinase

Reviewers' comments:

Reviewer #1 (Remarks to the Author):

In this manuscript the authors show that glucokinase is a *L*-Arginine-specific target protein. In Nit-1 beta cells, three glucokinase mutants (E256, E442 and E2443) were identified to be relevant for *L*-Arginine binding and *L*-Arginine-induced insulin secretion. In addition, it is demonstrated that *L*-Arginine prevents ubiquitination-dependent proteolysis of glucokinase by the chemical inducible E3 ubiquitin ligase cereblon. The authors also give a link to a MODY2 patient carrying E442 mutation in the glucokinase showing lower insulin secretion

in response to *L*-Arginine.

The manuscript deals with an interesting and novel research question. The topic is considered as of importance for its field, however, there are several major concerns of the manuscript, that make it difficult for publication in its current form.

Major points:

1. Most experiments have been repeated for only three times. For immunoblots the numbers of independent experiments is not mentioned. According to the standards of the British Journal of Pharmacology five is a minimum of numbers needed for reliable statistics (Curtis, Alexander et al. 2018). The authors have to increase their n numbers throughout the whole manuscript, human studies might be excluded due to the lack of patients. In addition, with reference to several guide lines it is more transparent to just plot the independent data points for the different measurements (Vaux 2012). The authors have to change this throughout the manuscript.

* First, we would like to express our gratitude for a number of valuable comments. We increased the sample number to n=6 for key experiments. Including experiments we performed for setting conditions, total number of experiments exceed 10 times with similar results. Also, we employed multiple approach to confirm one result. For example, to measure G6P, we used G6P ELISA, metabolome, and ADP ELISA. Our key findings were tested three ways by in vitro analysis, cell culture analysis, and MODY patients. We would like to stress that we got similar results.

2. The manuscript needs critical reading. Several sentences in the Manuscript and the Figure legends are incorrect or words appear twice (e.g. Glucokinase is interacts with arginine in beta cells – beginning of the results part; Figure 2c In the presence of glucose, *L*-Arginine more stimulates insulin secretion more than *D*-arginine).

* I apologize for grammatical issues. I made necessary correction to the manuscript.

3. In some Figures the unit for secreted insulin is not given. Please indicate the unit of the measured insulin (e.g. pg/cell/min) or if and how the data has been normalized (e.g. Figure 1a,b, Figure 2b).

* Thank you for your good suggestion. I updated units of secreted insulin unit.

4. In many figures it is a challenge for the reader to recognize what exactly was measured. The axis labeling is often lost in the legends of the curves presented in the figures. The axis labeling should be clearly visible (e.g. Figure 1a,b). The labeling for curve legends should be shown above or besides the curves or columns but not along the y-axis labeling.

* Thank you for your good suggestion. I changed y-axis labeling as recommended.

Minor points:

1. In figure legend 3 last sentence: f and g do not exist.

* Thank you for your good suggestion. I deleted it.

2. Why are p-values and asterisk mentioned in some figures (Figure 2c,d). It is difficult to see which comparisons shall be indicated with the individual p-values.

* Thank you for your good suggestion. I deleted p-values.

3. Spelling should be consistent throughout the manuscript L-Arginine or L-arginine

* Thank you for your good suggestion. "L-arginine" is now used throughout except in the beginning of sentences where "L-Arginine" is used.

Curtis, M. J., S. Alexander, G. Cirino, J. R. Docherty, C. H. George, M. A. Gienbycz, D. Hoyer, P. A. Insel, A. A. Izzo, Y. Ji, D. J. MacEwan, C. G. Sobey, S. C. Stanford, M. M. Teixeira, S. Wonnacott and A. Ahluwalia (2018). "Experimental design and analysis and their reporting II: updated and simplified guidance for authors and peer reviewers." *Br J Pharmacol* 175(7): 987-993.

Vaux, D. L. (2012). "Know when your numbers are significant." *Nature* 492(7428): 180-181.

Reviewer #2 (Remarks to the Author):

The hypothesis tested in this study is good and interesting but there are several improvements that have to be made in this manuscript. It would be fruitful for this manuscript to improved discussion with more authentic data rather than citing unpublished (submitted) manuscripts/data. It is also encouraged to cite relevant literature while writing methods section for authenticity of your research work. It would be highly appreciable, if two pics (f and g) from Fig. S3 be included in the main manuscript. Supplementary information Fig. S4 has not been referred anywhere in the manuscript. Please see the attached file for details.

* Thank you for your good suggestion. I deleted p-values.

Reviewer #3 (Remarks to the Author):

In this study the authors describe that *L*-arginine (2 mM) increases glucose-6-phosphate generation and glucokinase activity as well as glucose-induced insulin secretion through interaction and activation of glucokinase in the NIT-1 beta cell line. They propose that this effect is achieved through interaction of *L*-arginine with three specific amino acids in the glucokinase protein. This effect is absent in a glucokinase mutant which causes in patients MODY2.

General comment: The effects are of interest but they require further experiments for purposes of support as detailed below.

However, it is very irritating that the authors have submitted a manuscript which is rather a draft but not a manuscript in a proper final form. In many locations in the text on pages 14-15, there are still “XXX” symbols in the text rather than proper information. In addition the English language requires intensive revision.

* We would like to express our gratitude to the reviewer for his/her valuable comments and thorough review and for considering merits of our study. We have carefully read the comments/suggestions; and in response to those comments, we have revised the manuscript accordingly. I do apologize for using XXX. I am afraid I cannot provide specific names because this is a blind form. However, information is disclosed in the cover page and will be published for the accepted manuscript.

Specific comments:

1. Additional information and support must be through the following additional experiments, to provide additional support for their conclusions:

a) Measurement of glucose-dependent glucokinase activity in the absence and presence of 1 mM and 2 mM *L*-arginine *in vitro* with isolated recombinant human glucokinase.

* Thank you for your good suggestion. Glucose-6-phosphate production was measured using recombinant glucokinase protein expressed by *E. coli*. This is shown in Fig. 5h. The data suggested that arginine stimulates glucokinase activity *in vitro*.

b) Identical experiments with a cytosolic glucokinase extract from NIT-1 cells.

* Thank you for your good suggestion. When we performed similar experiments using NIT1 extract, we could not get similar results, which could be due to low level of expression reducing sensitivity of detection. So, we transfected HEK293FT cells which allow high level of expression and could demonstrate that *L*-arginine stimulates glucokinase activity *in vitro*. (Supplemental Fig. S5e).

2. Mention the arginine concentration at the beginning of the results text and in the legends to the figures.

* Thank you for your good suggestion. The concentration of arginine utilized was added as requested.

3. Explain “cereblon” in more detail and with reference.

* Thank you for your good suggestion. In the introduction section, I added description of cereblon with references.

4. It is well-known that beta cell glucokinase enzyme activity is increased through activation by interaction with PFK2 (such aspects of other established activators of glucokinase are not mentioned in this MS). Therefore the question: might arginine increase glucokinase activity in some way through interaction with the binding of PFK2 to glucokinase?

* Thank you for your good suggestion. In our previous paper, we isolated HK1, PFK1 and PFK2 as arginine-binding protein from HeLa cells. From NIT-1 cell extract, we could isolated glucokinase only without PFKs suggesting that PFK2 expression or affinity to arginine/glucokinase is lower in NIT-1 cells. In fact, glucokinase and PFK2 were overexpressed in the paper below, which may explain the lack of interaction with PFK2 in our system.

Laura Massa, Simone Baltrusch, David A Okar, Alex J Lange, Sigurd Lenzen, Markus Tiedge, Interaction of 6-phosphofructo-2-kinase/fructose-2,6-bisphosphatase (PFK-2/FBPase-2) With Glucokinase Activates Glucose Phosphorylation and Glucose Metabolism in Insulin-Producing Cells, *Diabetes* 53:1020-9 (2004). doi: 10.2337/diabetes.53.4.1020.

5. In liver GRP (glucokinase regulatory protein) is a strong inhibitor of GK activity. Is there evidence for interaction of arginine with GRP binding to GK?

* Thank you for your good suggestion. In general, GRP is highly expressed in the liver, but not in the pancreatic β -cells. When we have tested HepG2 cell extracts using new arginine-binding protein (unpublished result), MS data did not show GRP indicating that interaction between GRP and arginine is low.

6. Fig. 1: It is irritating and cannot be explained why the authors observed in Fig. 1b, that G6P but not glucose induced insulin secretion from NIT-1 cells. Rather the opposite should be the case, because glucose but not the phosphorylated G6P can enter an intact cell. This result is completely implausible.

* Thank you for your good suggestion. We were very much perplexed when observed insulin secretion by addition of G6P (Fig 1b). To confirm our data, we analyzed intracellular concentration of arginine, glucose, and G6P by metabolome analysis [ref26, 27] and obtained followings (new Supplemental Fig. S1a and b). In the absence of arginine, glucose administration (Fig 1a) induces 4-fold increase of glucose but only 1.5-fold increase in G6P. In the presence of arginine, glucose administration (Fig 1a) induces 4-fold increase in glucose and 3.5-fold increase in G6P. In the absence and presence of arginine, G6P administration (Fig 1b) induces 6-fold increase in G6P but no change in glucose. These data suggested that exogenous G6P can increase intracellular G6P directly.

7. Can arginine immobilized with magnetic beads enter intact cells? Where is this documented?

* No, we do not expect magnetic beads to enter inside of the cells as it is similar to Dynabeads (Thermofisher). We used magnetic beads with arginine in cell free system as explained in Methods and our previous papers [ref 13, 25].

8. It is not correct when the authors state in legend to Fig. 1 that GK is localized in secretory vesicles. This statement completely obscure. The reviewer does not understand how such a statement in view of the general knowledge about GK location in the literature can be made.

* Please refer to publications below. We also showed localization of glucokinase to ER is limited (Fig. 1f).

Catherine Arden, Andrew Harbottle, Simone Baltrusch, Markus Tiedge, Lorraine Agius, Glucokinase Is an Integral Component of the Insulin Granules in Glucose-Responsive Insulin Secretory Cells and Does Not Translocate During Glucose Stimulation, Diabetes 2004 Sep;53(9):2346-52. doi: 10.2337/diabetes.53.9.2346.

Reviewer Assessment Report

Manuscript#: COMMSBIO-20-0648-T

Manuscript Title: *L*-Arginine releases glucokinase from cereblon-mediated ubiquitination and stimulates glucose-6-phosphate production in pancreatic beta cells

Comments/Recommendations

In this manuscript, authors proposed a pathway of *L*-arginine-induced insulin secretion in pancreatic β -cells. In which, authors did so by investigating following hypothesis that *L*-arginine binds to glucokinase, increases its activity and prevents from cereblon-mediated ubiquitination.

Some major comments are below:

Comment 1: page 3, paragraph # 2, line # 8: regarding this line “Indeed, little is known how the activity of glucokinase is regulated in pancreatic beta cells.” Following papers have extensively reviewed the regulation of glucokinase in pancreatic β -cells. In my opinion, it would better to take following literature in consideration and appropriate changes are should be made for improvement.

1. Matschinsky, F. M. & Wilson, D. F., (2019). The central role of glucokinase in glucose homeostasis: A perspective 50 years after demonstrating the presence of the enzyme in islets of Langerhans. *Frontiers in Physiology*, 10, 148.
2. Matschinsky, F. M. (2002). Regulation of pancreatic β -cell glucokinase: from basics to therapeutics. *Diabetes*, 51(suppl 3), S394-S404.
3. Baltrusch, S., & Tiedge, M. (2006). Glucokinase regulatory network in pancreatic β -cells and liver. *Diabetes*, 55(Supplement 2), S55-S64.

* We would like thank the reviewer for valuable comments that help to improve our manuscript tremendously. I added three references and one from reference 2 below.

4. Mahalingam, B., Cuesta-Munoz, A., Davis, E. A., Matschinsky, F. M., Harrison, R. W., Weber, I. T., Structural model of human glucokinase in complex with glucose and ATP: Implications for the mutants that cause hypo- and hyperglycemia, *Diabetes* 48:1698-705 (1999). doi: 10.2337/diabetes.48.9.1698.

Comment 2: page 5, paragraph # 2, line # 4-6 & 8-10:

1. Interestingly, induction of insulin secretion is faster after *L*-arginine administration compared with *D*-arginine as can be seen as difference in insulin secretion at 0.5-1.5 min at 7 mmol/l glucose (Fig. 2b).
2. Glucokinase bound to *L*-Arginine was eluted by *L*-arginine but not by *D*-arginine indicating that only *L*-arginine interacts with glucokinase (Fig. 2e).

Above both statements contradict with each other, one is showing *D*-arginine has partial or little interaction with GCK and second is indicating that only *L*-arginine interacts with GCK. why is it so?

*Thank you for your good suggestion. While stimulation of insulin secretion through stimulation of glucokinase activity we report here is specific for *L*-arginine, we published the manuscript that demonstrated *D*-arginine stimulates insulin secretion through its interaction with UGGT1, not through GCK [ref 25]. And Wajngot [ref 29] also reported that *L*-arginine and *D*-arginine induce insulin secretion through the common cascade and a stereo specific pathway.

Comment 3: page 6, paragraph # 1, line # 1-4: five mutants of glucokinase (Y214C, E256K, A456V, E157T, and E443Δ mainly from MODY 2 patient mutation) These mutations could not be found anywhere reported in the literature cited by author. Proper protocol-reference(s) is also not mentioned anywhere in the manuscript for mutant preparation.

*Thank you for your good suggestion. Some mutants are referenced from Cho [ref 25]. Glutamate residue is most acidic amino acids and a likely target of arginine that is most cationic amino acids. In support, we found UGGT1 binds to *L*- and *D*-arginine through its C-terminal region that has five glutamates. Thus, we mutated glutamate residues in glucokinase. Reference of other mutants is [ref 11].

Comment 4: page 10, paragraph # 3, line # 4-10: “Additionally, our other study showed that *L*- and *D*-arginine has a shared target in the ER for regulating insulin secretion (manuscript submitted). Here, we showed evidence of *L*-arginine-preferential signaling through glucokinase in the secretory vesicles. Thus, arginine does appear to act via at least two signaling pathways, one responsive to both *L*- and *D*- arginine in the ER and the other only responsive to *L*-arginine at the secretory granules, both synergistically act to stimulate insulin secretion.”

In this paragraph, authors have suggested two pathways for *L*-arginine working, as authors have cited their own unpublished manuscript here. There are not enough proves mentioned in this manuscript for two pathways of *L*-arginine working. Please support your hypothesis with proofs either in the form of supplementary information or by citing appropriate literature.

Thank you for your good suggestion. The manuscript mentioned is now published as ref 25. Please kindly refer to our response to **Comment 2 as well. We estimate that *L*- and *D*-arginine binds to UGGT1, and UGGT1 regulates approximately 90% intracellular insulin in the ER. Only *L*-arginine binds to glucokinase, and glucokinase regulates approximately 10% intracellular insulin.

Comment 5: page 11-14: Methods

1. Insulin secretion analysis: protocol is difficult to understand, please rephrase. Papers are not cited properly for this section and are irrelevant as well.
2. Imaging study: The protocol is not described well; please explain if you have employed

modified protocol from the literature that is cited in this section or not? Which imaging technology and machine was used; what software and other experimental conditions were followed?

*Thank you for your good suggestion. Please see below.

3. Glucokinase expression in HEK293 cell and so on: Not all the sections of methods are cited and if cited, not properly. Please address these issue. Remove the figure references from text, heading from the methods section.
4. Statistical analyses: what kind of softwares were used and types of statistical analyses were followed? For One-way/two-way ANOVA, what kind of test has been used single-sided or double sided? Information about the type of tests used to analyze data are not given in figure legends as well.

**Thank you for your good suggestion. We used three softwares of Excel, Excel statics (SSRI, Tokyo Japan) and STATVIEW. We first performed student-t test using Excel and Excel statics. If the $p < 0.05$ by Excel and Excel statics, we performed one-way/two way of ANOVA with Excel statics and STATVIEW.

Comment 6: page 9-11: Discussion: the discussion section is not well cited with literature and needs to be improved by with authentic information available elsewhere. And the last paragraph of discussion, summary, needs to be elaborated more.

*Thank you for your good suggestion. We revised discussion as recommended.

Comment 7: page 16-19: Bibliography: Reference style is not properly followed though out the bibliographic section. Year is not in its proper place, journal names are bold and used as abbreviated names and full names as well. Please follow the proper reference style.

*Thank you for your good suggestion. We changed reference style.

Some minor comments are below:

Comment 1: Typographical, grammatical mistakes: A lot of typographical mistakes are present, here are some identified in this manuscript.

1. It is would be better if authors use pancreatic β -cells instead of beta cells throughout the manuscript.

*Thank you for your suggestion, I changed to β -cells.

2. It is would be better if authors write with one kind of notation for *L*-arginine instead of *L*-Arginine/*L*-arginine throughout the manuscript. And *L*-arginine or arginine throughout the manuscript.

*Thank you for your suggestion. In the beginning of sentences; “*L*-Arginine”; in the sentences “*L*-arginine”.

3. If, is it possible for authors to write “stereoisomer” for *L*-arginine/*D*-arginine instead of

“isomer” throughout the manuscript.

* Thank you for your suggestion, I changed to stereoisomer.

4. If, is it possible for authors to write “partial binding to glucokinase” in abstract instead of “almost no binding”.

* Thank you for your suggestion. As D-arginine did not show binding activity in Fig. 2e, we have revised to “did not have measurable”.

5. “WT” means wild-type or something else? Please specify here in abstract or in manuscript where applicable.

* Thank you for your suggestion. We note abbreviation now.

6. Page 1, paragraph# 2, line # 11-13: if is it possible to use less glucokinase in this sentence. Please rephrase as “Lastly, we elucidated that the binding of *L*-arginine to glucokinase increases its activity by protecting it from E3 ubiquitin ligase-cereblon mediated ubiquitination and degradation.”

* Thank you for your suggestion. I changed the paragraph.

7. If, is it possible for authors to write single notation for glucokinase, (GCK or HK4) instead of both throughout the manuscript.

*Thank you for your suggestion. I have deleted HK4 and used only GCK when notation is used to describe (GCK^{E256K}).

8. Page 3, paragraph #1, last line: glucose-induced insulin secretion is denoted as (GSIS) that is glucose-stimulated insulin secretion (GSIS). Please make appropriate changes throughout the manuscript, if it is not wrong.

*Thank you for your suggestion. We made correction.

9. Page 4, paragraph #1, line #1: please follow correct citation style “whatever11,12” instead of “whatever11&12” throughout the manuscript.

*Thank you for your suggestion. I change to the reference style using &.

10. Page 5, paragraph #2, line #1: change “collective” to “collectively”.

*Thank you for your suggestion. I changed it.

11. Page 5, paragraph #2, last line: please change if possible “which is likely due to another binding protein designated as” to “which is likely due to arginine/proinsulin binding protein (A/PBP) that binds to both L- and D-arginine.”

*Thank you for your good suggestion. Recently this manuscript is published [ref 25]. Please refer to ref 6, like Comment 2 and 5.

12. Page 6, paragraph #1, line # 6-7: it seems that “Supplementary information Fig. S2a, b”

is wrongly cited here instead of “Supplementary information Fig. S3a, b, and c.” Please confirm this.

*Thank you for your suggestion. I changed the manuscript.

13. Page 6, paragraph #2, last line: it seems that “Supplementary information Fig. S2d and e” is wrongly cited here as supplementary file does not contain any Fig. S2d and e.”

*Thank you for your suggestion. I changed the manuscript.

14. Page 7, paragraph #1, line # 4: please write as “the increase in G6P by *L*-arginine (Fig. 3d), and arginine-induced insulin secretion (Fig. 3e).”

*Thank you for your suggestion. I changed the manuscript.

15. Page 7, paragraph #1, line # 11-14: please rephrase the last part of this sentence.

*Thank you for your suggestion. I changed the manuscript.

16. Page 7, paragraph #3, line # 3: write “motif (UIM^{21&22}) and glucokinase^{23&24}” as “motif (UIM)^{21,22} and glucokinase^{23,24}.” See point # 9.

*Thank you for your suggestion. I changed them.

17. Page 8, line # 1: remove “, (Fig. 5a),” from “arginine-depletion, (Fig. 5a),”.

*Thank you for your suggestion. I changed manuscript.

18. Page 8, line # 14-15: Break the paragraph into two paragraphs and add question mark “?” at the end of question.

*Thank you for your suggestion. I changed the manuscript.

19. Page 8, line # 15-19: please rephrase the sentence, it is difficult to understand by reading one time. What does 25 means in that sentence?

*Thank you for your suggestion. I changed the manuscript, and 25 is reference number.

20. Page 9, discussion, paragraph #1: supplementary information is wrongly cited throughout this paragraph and not present in supplementary file. Last sentence should be rephrased if possible; because it contains glucokinase word four times, that makes it sound gluco-gluco-gluco-gluco.

*Thank you for your suggestion. I changed the manuscript.

21. Page 10, paragraph #1, line #2: change “complex better that glucose...” to “complex better than glucose....”

*Thank you for your suggestion. I changed the manuscript.

22. Page 10, paragraph #2, line #4-5: it seems “Supplementary information Fig. S2a and c” is mistakenly cited here. Please confirm, if not, then please explain how?

*Thank you for your suggestion. I deleted these Fig. S2.

23. Page 11, line #1: please cite correctly “motif (UIM21-24)” as “motif (UIM)21-24”. See point #9.

*Thank you for your suggestion. I changed references number style.

24. Page 12, line #2: See point #9.

*Thank you for your suggestion. I changed as #9.

25. Page 13, paragraph # 3, line #2: See point #9.

*Thank you for your suggestion. I changed.

26. Page 14, paragraph # 1, last line: See point #9.

*Thank you for your suggestion. I changed.

Comment 2: Page 4-15, Results, Discussions and Methods: please correct sub-headings and remove periods from the last words of sub-headings, where applicable.

*Thank you for your suggestion. I changed subheadings.

Comment 3: Page 4-15, Results, Discussions and Methods: please write “Supplementary Fig. xxx” rather than “Supplementary Information Fig. xxx”.

*Thank you for your suggestion. I changed to “Supplemental Fig SX”.

Comments for supplementary information file

Comment 1: It is suggested that if supplementary information Fig. S3 f & g, be included in the main manuscript rather than supplementary data; it would be helpful for reader to easily understand the binding site of *L*-arginine by having glance over the structural visuals of glucokinase.

*Thank you for your suggestion. I changed the structure data to Fig. 3c and d.

Comment 2: Page 2, supplementary information, abbreviations: Add WT for wild-type, if applicable.

*Thank you for your suggestion. I put WT to abbreviate.

Comment 3: Page 3-7: remove the word “information” from “Supplementary Information Fig. xxx” throughout the supplementary file and manuscript as well.

*Thank you for your suggestion. I changed to “Supplemental Fig SX”.

Comment 4: throughout the manuscript and supplementary file: under the legends of all figures, please mention statistical tests applied and type of tests used e.g., single-sided or double-sided student’s t-test or one-way/two-way ANOVA or any other where applicable.

*Thank you for your suggestion. I changed the manuscript.

REVIEWERS' COMMENTS:

Reviewer #1 (Remarks to the Author):

The revised version of the manuscript « L-Arginine prevents cereblon-mediated ubiquitination of glucokinase and stimulates glucose-6-phosphate production in pancreatic β -cells » answers to most but not all major comments raised by the different reviewers.

Major point

The main criticism remains that there is no significant improvement of the English throughout the manuscript. Many sentences, already in the abstract and especially in the Figure Legends, have not been read carefully, words are duplicated or sentences are incomplete. A point of criticism that has also been noted by other reviewers. Only a few of the incorrect sentences are listed below as examples.

Main text:

"L-Arginine stimulated **production glucose-6-phosphate** (G6P), and induced insulin secretion."

"We used mutation analysis of glucokinase and identified **three glutamate residues** that mediates **binding** to L-arginine."

Figure legends:

"In the presence of glucose, L-arginine **more stimulated insulin secretion more** than D-arginine. In the absence of glucose, both L-and D-arginine stimulated insulin secretion similarly."

"Insulin secretion in response to L-arginine in NIT-1 cells expressing various glucokinase mutants showed **that reduction in insulin secretion** when E256K, E443 Δ , and A456V-mutants were expressed."

"Homeostatic model assessment of β -cell **(HOMA β ,** a), and C-peptide-to-glucose ratio (b) **of a** were reduced in subject with E442* glucokinase mutation were reduced **compared** with two controls with WT glucokinase."

Reviewer #3 (Remarks to the Author):

The MS has been improved through the revision.

The author has cited additionally three reviews.

However, two are old and one newer does not addresss the question of glucokinase regulation in detail.

The reviewer proposeds therefore to cite instead of these reviews the following review:

Lenzen S. A fresh view of glycolysis and glucokinase regulation: history and current status. J Biol Chem 289, 12189-12194, 2014

REVIEWERS' COMMENTS:

Reviewer #1 (Remarks to the Author):

The revised version of the manuscript « *L*-Arginine prevents cereblon-mediated ubiquitination of glucokinase and stimulates glucose-6-phosphate production in pancreatic β -cells » answers to most but not all major comments raised by the different reviewers.

Major point

The main criticism remains that there is no significant improvement of the English throughout the manuscript. Many sentences, already in the abstract and especially in the Figure Legends, have not been read carefully, words are duplicated or sentences are incomplete. A point of criticism that has also been noted by other reviewers. Only a few of the incorrect sentences are listed below as examples.

Main text:

“*L*-Arginine stimulated production glucose-6-phosphate (G6P), and induced insulin secretion.”

“We used mutation analysis of glucokinase and identified three glutamate residues that mediates binding to *L*-arginine.”

Figure legends:

“In the presence of glucose, *L*-arginine more stimulated insulin secretion more than *D*-arginine. In the absence of glucose, both *L*-and *D*-arginine stimulated insulin secretion similarly.”

“Insulin secretion in response to *L*-arginine in NIT-1 cells expressing various glucokinase mutants showed that reduction in insulin secretion when E256K, E443 Δ , and A456V-mutants were expressed.”

“Homeostatic model assessment of β -cell (HOMA β , a), and C-peptide-to-glucose ratio (b) of a were reduced in subject with E442* glucokinase mutation were reduced compared with two controls with WT glucokinase.”

* The manuscript is now revised with a proof reading by a Native English speaker to correct for grammatical errors.

Reviewer #3 (Remarks to the Author):

The MS has been improved through the revision.

The author has cited additionally three reviews.

However, two are old and one newer does not addresss the question of glucokinase regulation in detail.

The reviewer proposes therefore to cite instead of these reviews the following review:

Lenzen S. A fresh view of glycolysis and glucokinase regulation: history and current status. J Biol Chem 289, 12189-12194, 2014

* Thank you very much for your suggestion. Above reference is now added as new reference 12.

New ref 12; Lenzen S, A fresh view of glycolysis and glucokinase regulation: history and current status, J. Biol. Chem. 289, 12189-94 (2014). doi: 10.1074/jbc.R114.557314.